# Characteristics of inflammatory response and repair after experimental blast lung injury in rats

Jürg Hamacher[1,2,3,4]*, Yalda Hadizamani [1,2], Hanno Huwer[5,6,7], Ueli Moehrlen[2,8], Lia Bally[9], Uz Stammberger[2,10], Albrecht Wendel[11], Rudolf Lucas [12,13,14]

1 Pneumology, Clinic for General Internal Medicine, Lindenhofspital, Bern, Switzerland, 2 Lungen-und Atmungsstiftung, Bern, Switzerland, 3 Medical Clinic V—Pneumology, Allergology, Intensive Care Medicine, and Environmental Medicine, Faculty of Medicine, Saarland University, University Medical Centre of the Saarland, Homburg, Germany, 4 Institute for Clinical & Experimental Surgery, Faculty of Medicine, Saarland University, Homburg, Germany, 5 Department of Cardiothoracic Surgery, Völklingen Heart Centre, Völklingen, Germany, 6 Department of Human Genetics, Saarland University, Homburg, Saar, Germany, 7 Department of Thoracic and Cardiovascular Surgery of the University Hospital of Saarland, Homburg, Saarland, Germany, 8 Pediatric Surgery, University Children's Hospital Zurich, Zurich, Switzerland, 9 Department of Diabetes, Endocrinology, Clinical Nutrition and Metabolism Inselspital, Bern University Hospital and University of Bern, Bern, Switzerland, 10 STM ClinMedRes Consulting, Basel, Switzerland, 11 Biochemical Pharmacology, Department of Biology, University of Konstanz, Konstanz, Germany, 12 Vascular Biology Center, Augusta University, Augusta, GA, United States of America, 13 Department of Pharmacology and Toxicology, Augusta University, Augusta, GA, United States of America, 14 Department of Medicine, Medical College of Georgia, Augusta University, Augusta, GA, United States of America

* hamacher@greenmail.ch

**Data Availability Statement:** All relevant data are within the paper and its supporting information files.

## Abstract

### Background and objectives

Blast-induced lung injury is associated with inflammatory, which are characterised by disruption of the alveolar-capillary barrier, haemorrhage, pulmonary infiltrateration causing oedema formation, pro-inflammatory cytokine and chemokine release, and anti-inflammatory counter-regulation. The objective of the current study was to define sequence of such alterations in with establishing blast-induced lung injury in rats using an advanced blast generator.

### Methods

Rats underwent a standardized blast wave trauma and were euthanised at defined time points. Non-traumatised animals served as sham controls. Obtained samples from bronchoalveolar lavage fluid (BALF) at each time-point were assessed for histology, leukocyte infiltration and cytokine/chemokine profile.

### Results

After blast lung injury, significant haemorrhage and neutrophil infiltration were observed. Similarly, protein accumulation, lactate dehydrogenase activity (LDH), alveolar eicosanoid release, matrix metalloproteinase (MMP)-2 and -9, pro-Inflammatory cytokines, including tumour necrosis factor (TNF) and interleukin (IL) -6 raised up. While declining in the level of

**Funding:** The author(s) received no specific funding for this work.

**Competing interests:** The work of YH, UM, and JH have been funded by Lungen- und Atmungsstiftung Bern, Switzerland which provides non-restricted financial support toward research on lung, respiratory and sleep-related respiratory diseases as well as related rehabilitation and lifestyle change issues such as exercise, smoking cessation, etc. JH is the chair of Lungen- und Atmungsstiftung Bern and his position did not influence the design of the study, the collection of the data, the analysis or interpreta-tion of the data, the decision to submit the manuscript for publication, or the writing of the manuscript and did not present any financial conflicts. Also, the work of JH was supported by a grant from the Deutsche Forschungsgemeinschaft (FOR 321/2-1; research group "Endogenous tissue injury: Mechanisms of autodestruction") and by the Herrmann Josef Schieffer Prize of the "Freunde des Universitätsklinikums Homburg e.V.". All remaining authors (HH, MM, LB, US, AW and RL) declare no potential financial or non-financial conflict of interest with the work presented here.

anti-inflammatory cytokine IL-10 occurred. Ultimately, pulmonary oedema developed that increased to its maximum level within the first 1.5 h, then recovered within 24 h.

## Conclusion

Using a stablished model, can facilitate the study of inflammatory response to blast lung injury. Following the blast injury, alteration in cytokine/chemokine profile and activity of cells in the alveolar space occurs, which eventuates in alveolar epithelial barrier dysfunction and oedema formation. Most of these parameters exhibit time-dependent return to their basal status that is an indication to resilience of lungs to blast-induced lung injury.

## Introduction

A blast wave includes a combination of the shock wave and the dynamic overpressure [1]. Primary blast injuries occur when the shock (stress) wave travels through the tissues, depositing energy, especially in organs with a gas-liquid interface [1]. Lungs, because of their special structure composed of air-filled alveoli harbouring a single layer of epithelial cells surrounded by delicate microvascular networks, represent susceptible organs to such a form of injury, the consequences of which are called blast lung injury [1]. The blast wave may cause instantaneous lung injury, characterised by alveolar capillary rupture and subsequent intrapulmonary haemorrhage and formation of pulmonary oedema [1]. If compared to lung contusions, to the much more extended and rather generalized physical impact of the blast wave to the thorax the pulmonary blast wave injury is usually a generalized acute lung injury causing alveolar ruptures at virtually the whole lung tissue. This is also evidenced by a characteristic bilateral, "butterfly" pattern in chest computer tomography, whereas lung contusions are usually unilateral, and more localized, as typically seen in vehicle accidents or falls from great heights [2].

The prognosis of trauma patients appears to be related to a post-traumatic immunologic imbalance status [3]. Systemic inflammatory response syndrome can occur when pro-inflammatory status is the dominant force. If the counter-regulatory and anti-inflammatory mediators predominate, immunosuppression ("immunoparalysis") with an inconclusive elimination of microorganisms and septic complications may follow the procedure [4]. A host inflammatory response is necessary to orchestrate tissue repair and immune competence following severe trauma.

Blast exposure could induce inflammation, oxidative stress, and cell apoptosis in the lungs of experimental animals and humans [5]. Accordingly, the inflammatory response has been defined as an imbalance of signalling molecules and the pro- or anti-inflammatory profiles of subjects. Inflammatory responses are believed to perform a key role in the development of blast exposure-induced lung injury [6]. Blast limb trauma causes remote lung injury, which is likely associated with a remarkable inflammatory response, oxidative stress, and the depletion of protective mechanisms. Multiple factors, including TNF, IL-6, IL-1β and albumin protein in the BALF exacerbate lung tissue fibrosis and respiratory dysfunction following lower extremity blast trauma in rats [7, 8]. A robust and selective response of CD43Lo/His48Hi monocytes and neutrophils in blood and lung tissue following primary blast lung injury occurs [8].

Regardless of whether the chest trauma is blunt or penetrating, the risk of alveolar haemorrhage caused by ruptured alveoli and haemorrhage around pulmonary vessels in blast wave victims must always be considered [9]. A principal reason for preventable death as a result of an explosion could be haemorrhage [1]. Within minutes to hours after a millisecond to

seconds of a blast wave, the alveolar wall ruptures with injury to the endothelial-based alveolar-interstitial and pneumocyte type I-dominated interstitial-alveolar membrane, causing scattered foci of alveolar bleeding, leading to accumulation of erythrocytes and proteinaceous blood serum deposits, and a much larger parenchymal laceration including emphysematous destruction or pneumothorax [10].

A comprehensive perception of the mechanisms underlying the cellular immune response to blast lung injury is of eminent importance, both in terms of disease prognosis and targeted futures in therapeutic aspects. Therefore, the current study, which involves establishing an experimental model of blast-induced lung injury in rats, aims to investigate the sequence of events as well as consecutive repair patterns over time and in comparison to non-blasted sham controls.

## Materials and methods

### 2.1 Animals

The investigation conformed to recommendations published in the Guide for the Care and Use of Laboratory Animals (NIH publication No. 85–23, revised 1996) and was in accordance with the German legislation on the protection of animals. In all experiments female Wistar rats (Weight 230 ± 20 g; Harlan Winkelman GmbH, Borchen, Germany) were used. The animals were housed in a central, specific-pathogen-free animal housing facility under the oversight of the institutional animal welfare officer. The animals were fed standard rat chow and had access to water ad libitum until the start of the experiment. The study was approved by the Federal Animal Care and Use Committee in Tübingen, Germany.

### 2.2 Blast wave generator and pressure wave monitoring

The blast wave-induced lung injury model consisted of a standardised discharge of a pressure wave at a defined time point of the respiratory cycle. The magnitude of the pressure wave measured at the thorax level was in the range of what causes blast lung injury in humans [11]. A focal blast wave generator was used to deliver a reproducible blast wave injury in a laboratory environment simulating free air burst blast pressures by a detonation. The blast wave generator was originally designed by Irwin and co-workers and modified by the Liener team [12, 13] to discharge a reproducible blast pressure wave in a laboratory environment. A blast wave generator (Department of Scientific Engineering and Services, University of Konstanz, Germany) was used based on our specific needs in order to improve the pressure wave monitoring and the reproducibility of pressure wave exposure.

In brief, the blast wave generator consisted of a pressure reservoir, which was separated from the blast nozzle by an airtightly fixed polyester film (Strohmeier, Rheda-Wiedenbrück, Germany) that separated the two compartments (Fig 1A). A compressed air bottle was loaded to 18 bar. The computer-triggered opening of an ultra-fast industrial valve (HEE-D-MINI-24, opening time 25 ms, Festo AG & CO, Esslingen, Germany) between the compressed air bottle and the pressure reservoir resulted in an abrupt pressure rise and subsequent burst of the polyester diaphragm. Thus, a blast pressure wave was discharged through the nozzle towards the rat. Pressure sensors were located in the pressure reservoir and on each side of the rat's thorax. An additional transducer recorded the breathing frequency and the respective breathing phase of the animal. Via this transducer, the valve was triggered automatically. The rats were anaesthetized in a bell jar flooded with a Halothane® (Sigma-Aldrich Chemie GmbH, Deisenhofen, Germany)/$O_2$ gas mixture through a Halothane-evaporator (flow: 2 L/min oxygen, 4% Halothane). Then the animals were placed on a pad and positioned in a supine position, with the paws fixed by tape (Fig 1B). Up to the pressure wave exposure, they were maintained via an

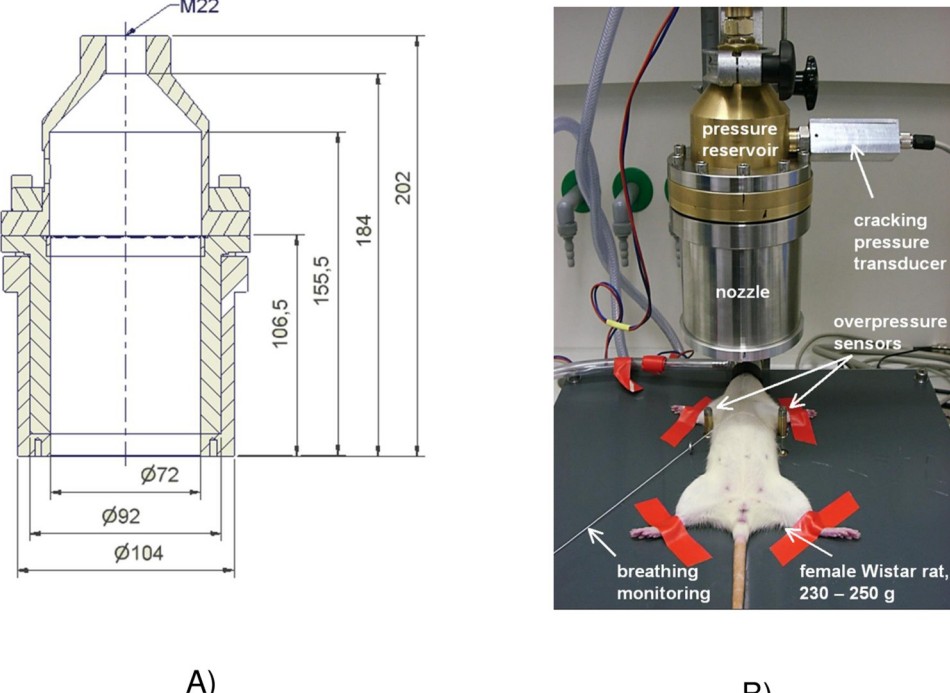

A)

B)

**Fig 1. A)** Schematic diagram of the blast wave generator (front view). The dimension is in millimetres (mm). **B)** The blast wave generator includes pressure transducers at different positions. A cracking pressure transducer in the pressure reservoir, two overpressure sensors on the left and right sides of the rat and a transducer for recording the breathing frequency.

anaesthesia mask (flow: 1.5 L/min oxygen, 4% halothane). In addition, the animals received the analgesic Buprenorphine (Temgesic, Indivior Eu Ltd., Berkshire, Great Britain) (0.03 mg/kg BW) subcutaneously. Triggered by the breathing sensor, the rats were subjected to the pressure wave at the end of a tidal volume expiration. The blast pressure wave was centred on the xiphoid, with the tip of the blast nozzle located at a distance of 3.5 cm. Restauration of spontaneous breathing was recorded for one minute via the breathing sensor. Immediately after blast exposure, the animals were terminally anaesthetized by an intraperitoneal injection of sodium pentobarbital (Narcoren®, 160 mg/kg BW).

## 2.3 Bronchoalveolar Lavage (BAL)

Animals were randomly chosen and euthanized 10 min, 1.5 h, 3 h, 6 h, 12 h, 24 h, and 96 h after trauma (S1 Fig); non-traumatized animals served as sham controls. Lungs were tracheotomized and gently rinsed with 6 ml of ice-cold NaCl 0.9% solution (Delta-Pharma Boehringer-Ingelheim, Germany). Recovery of the instillate was always > 50%. In order to obtain BAL fluid (BALF) supernatant, BALF was centrifuged at 340 x g for 8 min, and the supernatants were stored at -70˚C.

**2.3.1 Determination of the alveolar haemorrhage.** Pulmonary haemorrhage was quantified using the total erythrocyte count in the BALF. The erythrocyte number was assayed in a Neubauer counting chamber.

**2.3.2 Total and differential cell counts and cell viability.** To determine the total BAL cell count, BALF was mixed with Trypan Blue (Sigma-Aldrich Chemie GmbH, Deisenhofen, Germany) solution (1:2) and incubated for one minute at room temperature. The cell number was

assayed using a Neubauer counting chamber. Cells that excluded the Trypan dye were regarded as viable, while blue cells were considered dead.

Cytospin preparations were used to determine the BAL cell differential. Depending on the total cell counts the BAL samples were first diluted (2–4 fold) in isolation medium before pipetting 200 µl into the cytospin inserts. The samples were centrifuged at 90 x g (Minifuge RF; Heraeus Instruments, Hanau, Germany) for 7 min. The cells were fixed and stained with May-Grünwald/Giemsa solution (Merck KGaA, Darmstadt, Germany). From every experiment, two cytospins were made, and a digital photograph was taken (Nikon Coolpix 995 through a customised ocular with a Zeiss Axiovert 50). The numbers of alveolar macrophages (AMs), lymphocytes, and infiltrated polymorphonuclear leukocytes (PMN) were assessed by using standard morphological criteria for differentiation.

Total blood leukocytes were determined in the Neubauer counting chamber by mixing heparin blood (Liquemin® in NaCl, 1:10) with Türk's solution (1:10). For the differential blood count 10–15 µl whole blood were dispensed on a microscope slide, dried and stained with May-Grünwald/Giemsa solution.

**2.3.3 Determination of total protein content.**   The BAL supernatants were stored at -70°C. A standard 2 mg/ml bovine serum albumin (BSA) was diluted (2000, 1000, 500, 250, 125, 62.5, 31.25, 0 µg/ml). 10 µl sample and standard were added on a microtiter plate, respectively. 190 µl/well Pierce BCA protein assay reagent (Bender & Hobein GmbH, Heidelberg, Germany) was added and incubated for 30 min at 37°C. The extinction was measured at 550 nm in an ELISA reader. Protein content was calculated using the BSA standard curve.

**2.3.4 Measurement of lactate dehydrogenase (LDH).**   Lactate dehydrogenase (LDH) was measured in BALF according to the recommendations of the German Society for Clinical Chemistry. Measurement of LDH activity was performed in an Eppendorf ACP 5040 Analyser (Netheler & Hinz, Hamburg, Germany).

**2.3.5 Cytokine determination.**   For TNF measurements, samples were stored at –70°C and quantified by the sandwich enzyme-linked immunosorbent assay (ELISA). Antibody pairs (polyclonal rabbit anti-rat or anti-mouse) and recombinant TNF serving as standards were purchased from Pharmingen (San Diego, CA, USA). ELISA plates were coated overnight at 4°C with 50 µl/well with coat Ab (0.5% rabbit anti-TNF serum) diluted in coating buffer (1:200). After blocking with 200 µl/well of phosphate-buffered saline solution, PBS (PAA, Germany)/3% BSA at pH 7.0 for 2 h at room temperature, the plates were washed twice with PBS/ 0.05% (PAA, Germany) Tween 20 (Sigma-Aldrich Chemie GmbH, Deisenhofen, Germany). Samples and standard (50 µl/well) were added and incubated for 3 h. After four wash cycles, 100 µl/well 0.5 mg/ml tracer antibody in PBS/3% BSA were added and incubated for 45 minutes. After another six wash cycles, plates were incubated for 30 minutes with 100 µl/well 50 ng/ml streptavidin-peroxidase (Jackson Immuno Research, West Grove, PA, USA) in PBS/3% BSA. Following eight wash cycles, 100 µl/well TMB liquid substrate solution (Sigma, Deisenhofen, Germany) was added and incubated for 5–15 minutes. After addition of 50 µl/well stop solution (1 M H2SO4), absorption was measured at 450 nm using a reference wavelength of 690 nm in an ELISA reader. The detection limit of the assay was 10 pg/ml.

IL-10 and IL-6 were determined using a commercially available ELISA kit (Biosource Europe SA, Nivelles, Belgium). The detection limits of the assays were 15.6 pg/ml for IL-10 and 31 pg/ml for IL-6. Assay and analysis were performed according to the manufacturer's recommendations.

**2.3.6 Measurement of cytokine-induced neutrophil chemoattractant 3 (CINC-3).**   Samples taken from BAL were stored at -20°C. The chemokine CINC-3 was assessed with the rat DuoSet ELISA Kit (R&D Systems Europe, Abingdon, UK) and performed according to the manufacturer's recommendations.

**2.3.7 Measurement of eicosanoid (thromboxane A2, 6-keto-PGF1α) concentrations.**
BAL samples were stored at –70˚C. Thromboxane A2 (t1/2 = 30 sec) was assessed as the stable by-product thromboxane B2 (TXB2). Both TXB2 and prostacyclin (6-keto-PGF1α) were determined by an EIA kit (BioTrend GmbH, Köln, Germany) according to the manufacturer's instructions. The detection limits of the assays were 13.7 pg/ml TXB2 and 3.2 pg/ml 6-keto-PGF1α. The cross-reactivity of the detecting antibody was TXB2 100%, 2, 3—dinor TXB2 7.1%, prostaglandins<0.01% and 6-keto-PGF1α 100%, 2, 3-dinor 6-keto-PGF1α 3.17%, PGF2α 1.67%, prostaglandins 0.2–0.6%, respectively.

**2.3.8 Gelatin zymography.** To detect the gelatinolytic activity of MMP-2 and -9 in BALFs, the samples of 3 to 5 animal per time point were evaluated by gelatin zymography as described previously [14]. In brief, 300 μl BAL fluid was concentrated by centrifuge. The supernatant was subjected to sodium dodecyl sulfate-polyacrylamide gel electrophoresis. After electrophoresis, the gels were incubated, fixed, and stained with 0.2% Coomassie brilliant blue R250 (Serva, Heidelberg, Germany). The gels were decolorized, and gelatin digestion was identified.

As markers, we used the purified pro-matrix metalloproteinase-2 (1 ng MMP-2, isolated from human rheumatoid synovial fibroblasts; Merck KGaA, Darmstadt, Germany) and pro-matrix metalloproteinase-9 (0.1 ng MMP-9, isolated from human neutrophil granulocytes; Merck KGaA, Darmstadt, Germany).

## 2.4 Wet to dry ratio of lung tissue

After the animal was sacrificed, the lungs were excised "en bloc" and cleaned of non-lung tissue and blood. The lung wet weight was determined before drying the lung tissue in an oven at 50˚C. After three days, a stable dry weight of the lungs was achieved, and the wet to dry weight ratio was calculated.

## 2.5 Histological examinations

After thoracotomy and non-recirculating perfusion of the lung with PBS, the lung was excised, intratracheally infused with 4% buffered formalin (Sigma-Aldrich Chemie GmbH, Deisenhofen, Germany), and embedded in paraffin (Sigma-Aldrich Chemie GmbH, Deisenhofen, Germany). Five μm sections were stained with haematoxylin and eosin (Merck KGaA, Darmstadt, Germany) and examined by light microscopy.

## 2.6 Statistics

Data in the figures are given as means ± SEM, data in the tables as mean ± SD. Entire curves data from two experimental settings were compared by a two-way ANOVA design. Values of p<0.05 were considered statistically significant. Data from the end or other time points were analysed by one-way ANOVA. In case of differences among the groups, post-tests were performed as indicated in the legends (Dennett's multiple comparison test, Tukey's multiple comparison test, or Bonferroni's multiple comparison test); statistics were performed with the Graph Pad Prism 3.0 software (Graph Pad Software, San Diego, CA, USA). Of note, we have done all the controls needed to use the parametric tests.

## Results

### 3.1 Experimental timeline and blast model

Pressure curves over time measured by the transducers located on the right and left sides of the animal's chest at the distance of 3.5 cm of the blast source to the thorax revealed a mean

pressure peak of 3.16 ± 0.43 bar and a pressure wave duration of 630 ± 30 μs (n = 219). To improve reproducibility, animals were blasted in the same breathing state, i.e., during spontaneous tidal volume breathing under halothane anaesthesia at the end of expiration. Under these circumstances, overall mortality observed in all experiments was 14.5%., with most of the deaths appearing during the first minutes post-injury.

In the following sections, physiologic responses to thoracic trauma have been provided.

## 3.2 Lung blast injury temporarily disrupts the integrity of the alveolar-capillary barrier in rats

Within 10 minutes following blast pressure wave exposure, extensive alveolar haemorrhage and exudation of fluid into the interstitial and alveolar spaces were detectable, accompanied by a disruption of alveolar septae (Fig 2A). Three hours after blast exposure, there was a marked focal alveolar haemorrhage, while leukocyte infiltrates could not yet be detected (Fig 2B). Twenty four hours after the blast, erythrocyte-laden AMs were observed, accompanied by a few detectable PMN in the alveolar space (Fig 2C). Fig 2D and 2E represent the histological picture of the healthy normal lung tissue. Intra-alveolar haemorrhage, quantified by red blood cell counts (RBCs) in BALFs of traumatized rat lungs showed a significant increase within the first 10 minutes, but decreased after 6 h, in order to return to undetectable levels by 96 h after blast (Fig 3). As shown in Fig 4, a significant accumulation of protein occurred as early as 10 min following blast exposure in the alveolar space of traumatized lungs, as compared to non-traumatized controls (p<0.0001). After 24 h, protein levels decreased and 96 h following trauma protein concentration in BALF returned to control levels. The release of the cytosolic enzyme LDH into the BAL was used as a marker for cellular integrity following exposure to the blast. In conformity with the protein accumulation and RBC counts, LDH activity was remarkably elevated within ten min following pressure wave exposure (p<0.01) (Fig 5), but returned to control levels 96 h after the blast. Pulmonary injury, quantified by the lung wet to

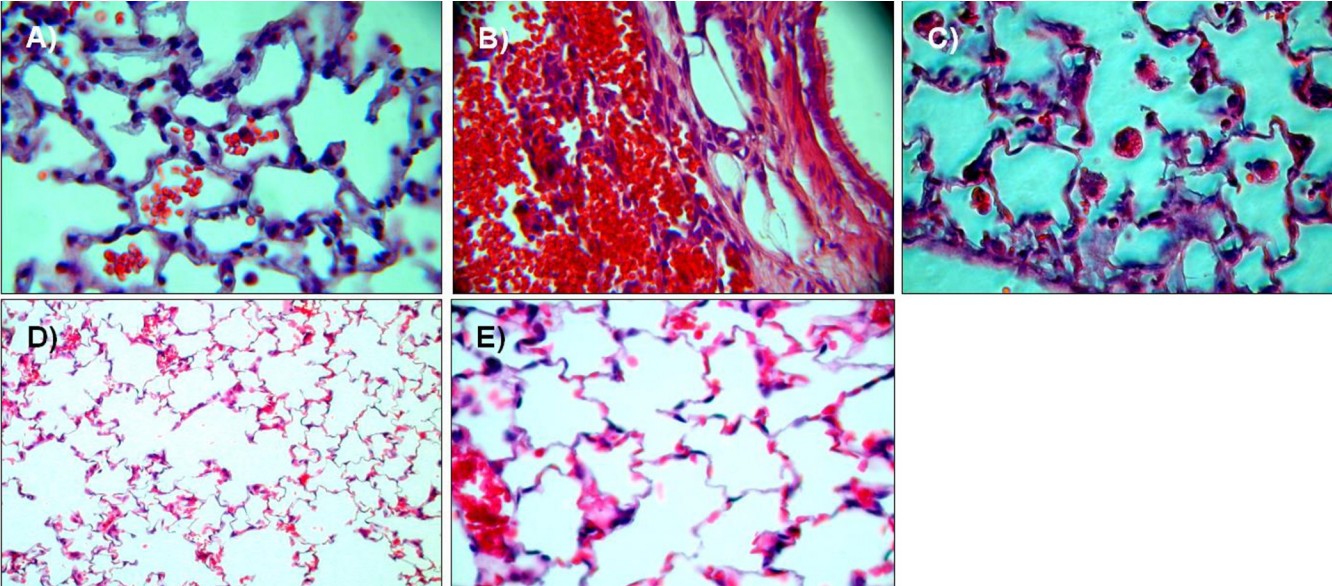

**Fig 2. Histopathology of rat blast injury.** Rats were exposed to the pressure wave at 3.5 cm from the nozzle. Animals were sacrificed either 10 min **A)**, 3 hours **B)** or 24 hours **C)** after blast exposure, non-traumatized animals served as sham controls **E)** + **D)**. Visualized at an original magnification of 40-fold **A)**–**C)** + **E)** and 10-fold in **D)**.

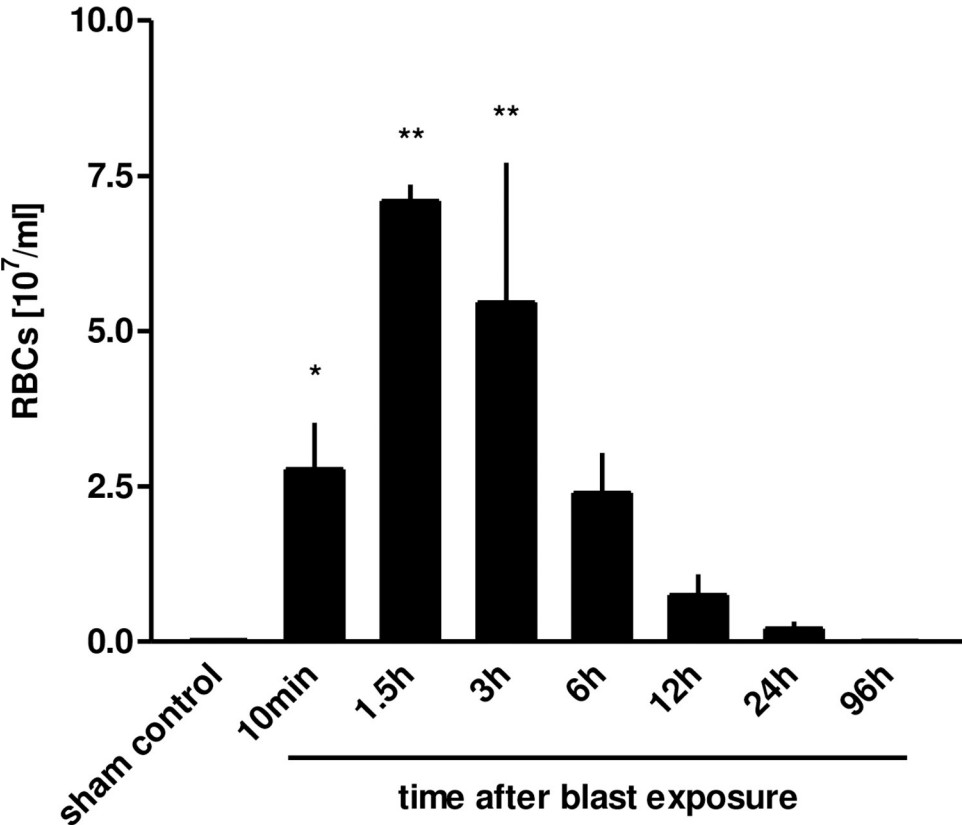

**Fig 3. Blast injury-related protein accumulation in BALFs.** Animals were sacrificed 10 min (n = 5), 1.5 h (n = 3), 3 h (n = 3), 6 h (n = 3), 12 h (n = 4), 24 h (n = 3) or 96 hours (n = 3) after blast exposure, non-traumatized animals served as sham controls (n = 9). Protein concentration was determined in the BALF. Data were analysed by one-way ANOVA and Dunnett's multiple comparison test: **p <0.01 vs. control. Data represent means ± SEM, number of individual experiments (n) in parentheses.

dry weight ratio (w/d lung weight ratio), was significantly enhanced within the initial 90 minutes following the blast (p<0.001), indicating pulmonary oedema formation At six hours the wet/dry ratio was significantly lower compared to 1.5 hours (p<0.001) (Fig 6).

### 3.3 Blast injury-induced release of inflammatory mediators

Since eicosanoids have been suggested to contribute to blast-related lung injury [15], we detected levels of thromboxane and prostacyclin in BALF at different time points following the blast. As shown in Fig 7A and 7B, a significant level of both thromboxane and prostacyclin was detected in BALF within 10 min ensuing blast pressure wave exposure (p<0.001), followed by a decrease by 90 min. At 96 h after blast exposure, we could no longer detect increased levels of these mediators.

### 3.4 Evolution of cytokine profile following blast wave exposure

Since both the inflammatory cytokines TNF and IL-6 have been shown to promote alveolar-capillary barrier dysfunction [16, 17], we examined their release into the BAL over time after blast injury. As shown in Fig 8A, increased TNF levels were detectable 3h after trauma (p<0.05 vs. control), reaching a maximum at 12h post blast exposure (p<0.01 vs. control). At later time-points, TNF levels continuously decreased (Fig 8A). In contrast, plasma TNF was

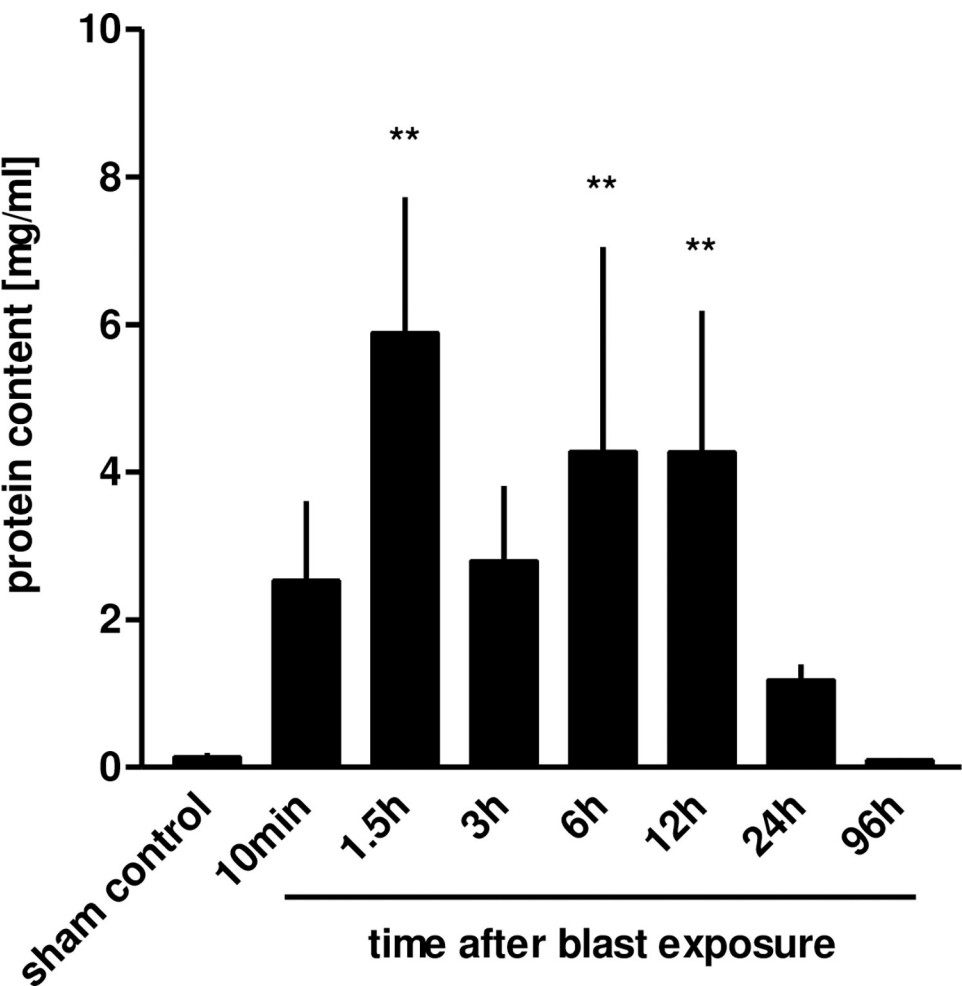

**Fig 4. Blast injury-related pulmonary haemorrhage.** The animals were sacrificed 10 min (n = 5), 1.5 h (n = 3), 3 h (n = 3), 6 h (n = 3), 12 h (n = 3), 24 h (n = 3) or 96 hours (n = 3) after blast exposure, non-traumatized animals served as sham controls (n = 8). RBCs were determined in BALFs. Data were analysed by one-way ANOVA and Dunnett's multiple comparison test: *p <0.05 and **p <0.01 vs. control. Data are means ± SEM, number of experiments (n) in parentheses.

not detectable at any time point up to 96 h after trauma, neither by ELISA nor by the WEHI-bioassay. IL-6 levels rose within 3h after trauma and reached significant peak levels after six hours (p<0.01 vs. control) (Fig 8B).

Furthermore, IL-10 levels were measured in BALFs. In comparison to non-traumatized sham controls, the IL-10 levels in BALFs of traumatized animals were significantly decreased (p<0.01 and p<0.05, respectively) at any time-point within 24 h and increased to physiological levels/baseline 96 h after blast injury (Fig 9).

### 3.5 Release of soluble tumour necrosis factor receptors (sTNFR)

Because of the lack of detectable plasma TNF and the relatively moderate levels of TNF in the BAL after blast, we examined soluble tumour necrosis factor receptor type I (sTNFR-I) and soluble tumour necrosis factor receptor type II (sTNFR-II) levels in BALF in response to blast injury, in order to test whether the TNF is bound to its receptors and therefore not active and possibly not detectable. Patterns of BAL 55 kDa sTNFR-I and BAL 75 kDa sTNFR-II at serial

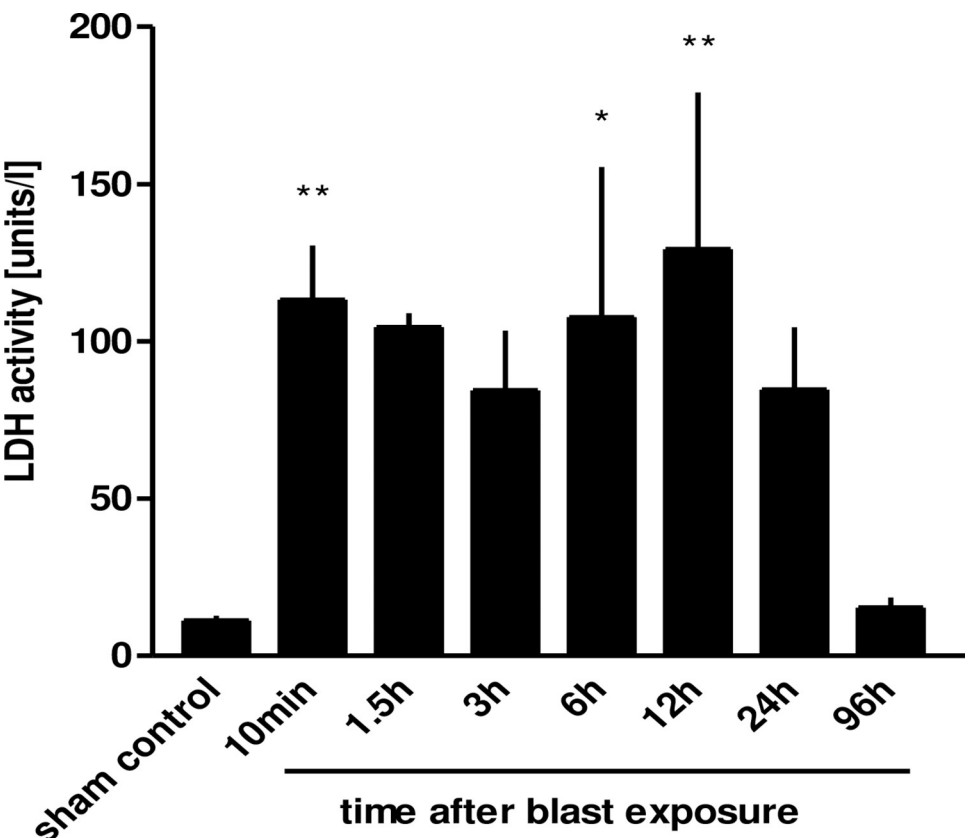

**Fig 5. LDH release after thoracic trauma.** The animals were sacrificed 10 min (n = 5), 1.5 h (n = 3), 3 h (n = 3), 6 h (n = 3), 12 h (n = 4), 24 h (n = 3) or 96 hours (n = 3) after blast exposure, non-traumatized animals served as sham controls (n = 8). LDH activity was determined in BALFs. Data were analysed by one-way ANOVA and Dunnett's multiple comparison test: *p <0.05 and **p <0.01 vs. control. Data are means ± SEM, number of experiments (n).

intervals after injury are depicted in Fig 10A. In comparison to controls, both receptors were significantly elevated 90 minutes after the blast (p<0.01). Thereafter, levels declined, followed by a second marked increase within the first 6 to 12 h (p<0.01), and reaching near baseline levels within 24 h.

Within the first 12 h after blast injury, there were no remarkable differences between sTNFR-I and sTNFR-II release. In contrast, 24 and 96 h after trauma the sTNFR-II levels were significantly higher than the sTNFR-I levels (p<0.01 and p<0.05, respectively). A comparison of the sTNFR-I and sTNFR-II levels of non-traumatized control lungs showed higher levels of sTNFR-II than of sTNFR-I (p<0.05). The sTNFR-I/II ratio was significantly elevated within 6 h of trauma compared to sham controls (p<0.05) and returned to reference ratios after 12 h (Fig 10B). A high correlation between both sTNFR subtypes could be shown over 96 h (r = 0.96; p<0.0001). The correlation between different detected plasma mediators revealed a correlation between both soluble TNF receptors and IL-6 levels over 96 h after blast exposure (r = 0.595; p<0.001 for sTNFR-I and r = 0.557; p<0.001 for sTNFR-II to IL-6, respectively).

### 3.6 Infiltration of inflammatory cells

**3.6.1 PMN infiltration.** Analysis of the measure of various cell types in lavage fluids extracted from traumatized groups isolated at different time-points shows an increased number of neutrophils in BAL and as such, PMN migration into the alveolar space starting at 90

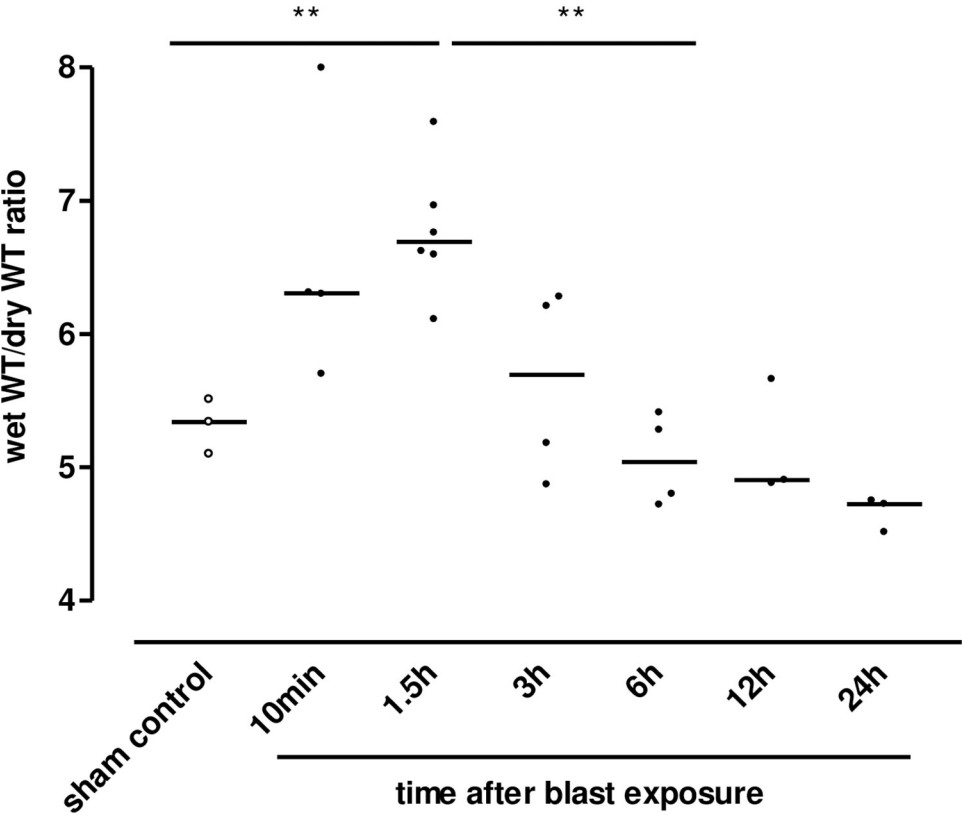

**Fig 6. Thorax trauma-induced weight gain.** Lung wet weight and dry weight were determined at different time-points after pressure wave exposure. The wet to dry weight ratios were calculated. Data were analysed by one-way ANOVA and Bonferroni's multiple comparison test: **p <0.01 1.5 h vs. sham control and 6 h after blast, respectively.

minutes following the blast. However, huge amounts of neutrophils were found not before 6 and 12 h after blast exposure (p<0.05 vs. sham controls) followed by a continuous decay within 96 h (Fig 11). The number of lymphocytes in the BALF did not vary significantly over time.

Fig 12A and 12B demonstrate the corresponding cytospins in the different BALFs. The cytospins of the non-traumatized sham controls exhibited a homogeneous population of AMs. Only very few lymphocytes could be found and no PMN (Fig 12A). At the time-point, 24 h post-blast injury, the PMNs showed about equal numbers compared to AMs. AMs were carrying high intracellular amounts of erythrocytes (Fig 12B). Quantitative analysis of peripheral blood leukocytes revealed no statistically significant differences within 96 h following blast exposure.

Furthermore, significant increased neutrophil related myeloperoxidase (MPO) activity was determined in BALF investigated 12 after blast exposure as compared to non-traumatized controls (p<0.01) (0.43 ± 0.24 relative units, n = 4; and 0.07 ± 0,004 relative units, n = 9). Table 1 summarizes the time-dependent results of BALFs investigated after blast injury and in sham controls.

**3.6.2 CINC-3 measurement in the BALFs.** To illuminate the performance of chemokines in thoracic trauma-induced infiltration of neutrophils, we quantified the α-chemokine CINC-3 (also called macrophage inflammatory protein-2, MIP-2), which represents one of three functional rat homologues to human Interleukin 8 (IL-8). The CINC-3 levels in the BALFs of traumatized rat lungs were augmented 6 h after trauma (p<0.01 vs. sham controls) and rebounded to the level of sham control after 96 h. The time course of the CINC-3 release correlated with the neutrophils' numbers (r = 0.63, p<0.001) (Fig 13).

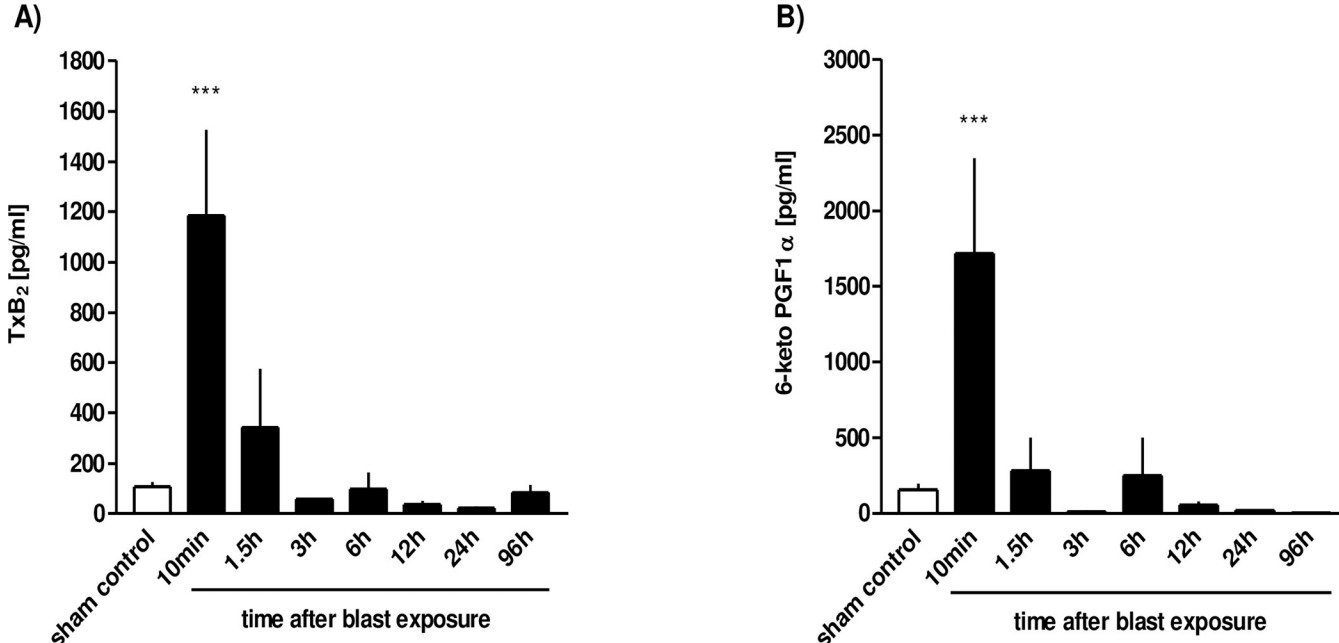

**Fig 7. Prostanoid release in the early phase after thoracic trauma.** The animals were sacrificed 10 min (n = 6), 1.5 h (n = 3), 3 h (n = 3), 6 h (n = 3), 12 h (n = 4), 24 h (n = 3) or 96 hours (n = 3) after blast exposure, non-traumatized animals served as sham controls (n = 9). **A)** TxB2 release and **B)** 6-keto PGF1α were determined in BALFs. Data were analysed by one-way ANOVA and Tukey's multiple comparison test. *** $p < 0.001$: 10min vs. all other groups. Data represent means ± SEM, number of experiments (n) in parentheses.

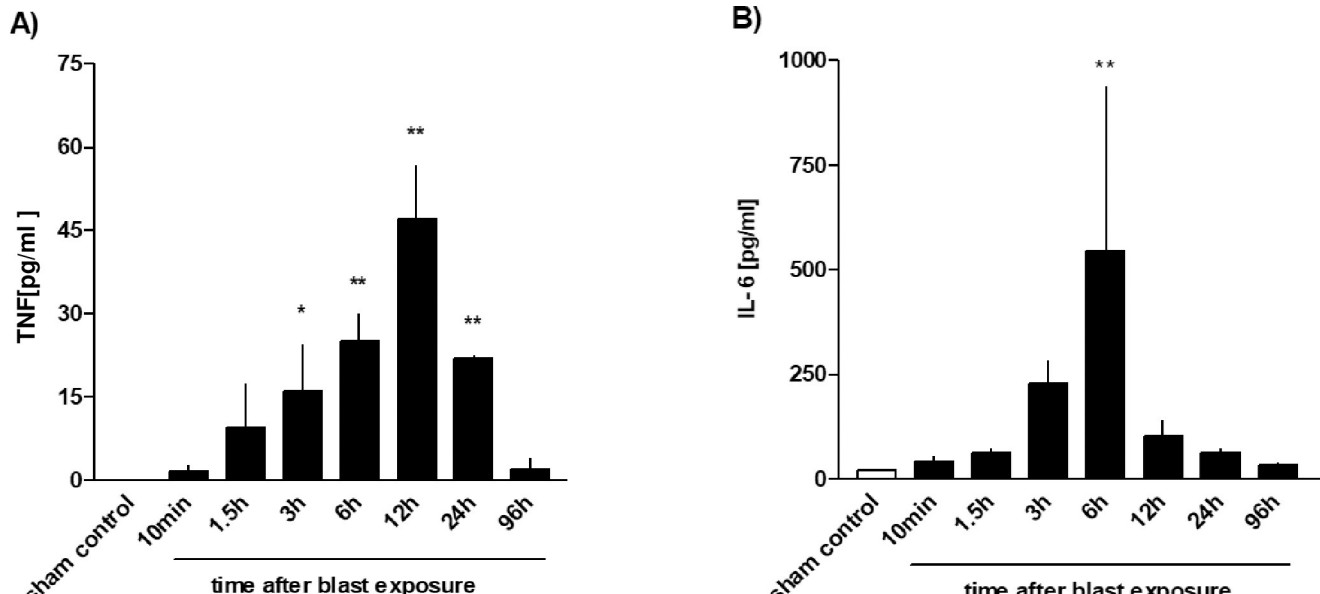

**Fig 8. Time course of TNF and IL-6 release after thoracic trauma.** The animals were sacrificed 10 min (n = 6), 1.5 h (n = 3), 3 h (n = 3), 6 h (n = 3), 12 h (n = 4), 24 h (n = 3) or 96 hours (n = 3) after blast exposure, non-traumatized animals served as sham controls (n = 9). **A)** TNF and **B)** IL-6 release were determined in BALFs. Data were statistically analysed by one-way ANOVA and Dunnett's multiple comparison test: *$p < 0.05$ and ** $p < 0.01$ vs. sham control. Data represent means ± SEM, number of experiments (n) in parentheses.

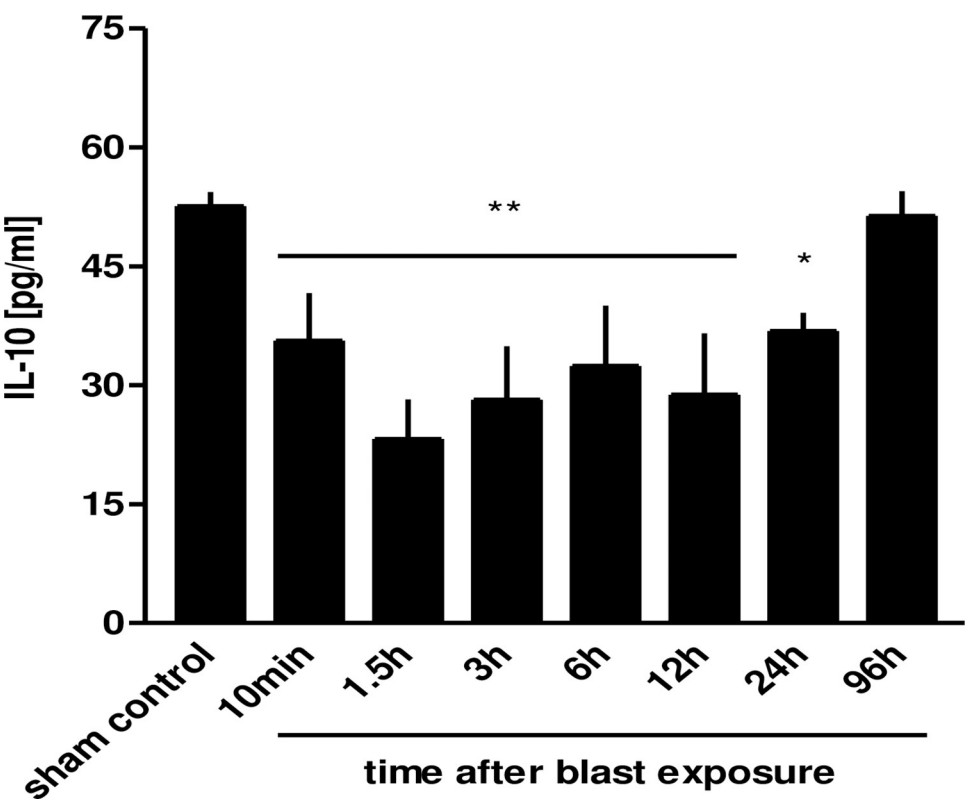

**Fig 9. Decrease in BAL IL-10 after thoracic trauma.** The animals were sacrificed 10 min (n = 6), 1.5 h (n = 3), 3 h (n = 3), 6 h (n = 3), 12 h (n = 4), 24 h (n = 3) or 96 hours (n = 3) after blast exposure, non-traumatized animals served as sham controls (n = 9). IL-10 release was assessed in BALFs. Data were statistically analysed by one-way ANOVA and Dunnett's multiple comparison test: *p<0.05 and **p<0.01 vs. sham control. Data are means ± SEM, number of experiments (n) in parentheses.

### 3.6.3 Matrix metalloproteinases in the BALF

According to the observed neutrophil infiltration as well as extracellular matrix remodelling, we investigated the release and activation of the MMP-2, (72 kDa type IV collagenase or gelatinase A) and MMP-9 (92 kDa type IV collagenase, 92 kDa gelatinase or gelatinase B) over time after thoracic trauma. Gelatinolytic bands of approximately 134, 92, 85, 72, and 66 kDa were detected in BALFs. The bands correspond to the pro-MMP-9 lipocalin complex, pro-MMP-9, the active form of MMP-9, pro-MMP-2, and active form of MMP-2, respectively [14]. Intense gelatinolytic bands corresponding to MMP-9 and coexpressed bands of the pro-MMP-9 lipocalin complex were discovered after 6 h, and weak bands after 12 and 24 h, whereas lytic bands equivalent to active MMP-9 were only barely detected at these time-points. In addition, a weak gelatinolytic band corresponding to active MMP-9 appeared initially after trauma.

MMP-9 activity after 6 h correlated with the augmented neutrophil count in BAL at this time-point. MMP-2 is reported to be preferentially released by fibroblasts and epithelial cells and may, therefore, be generated after trauma-induced cellular injury.

## Discussion

Blast thoracic injury seems to be the most dangerous type of non-penetrating thoracic injury that may more regularly lead to death due to acute lung injury than other non-penetrating thoracic injuries as in traffic accidents or by falls from a great height [2]. The lung, by virtue of its

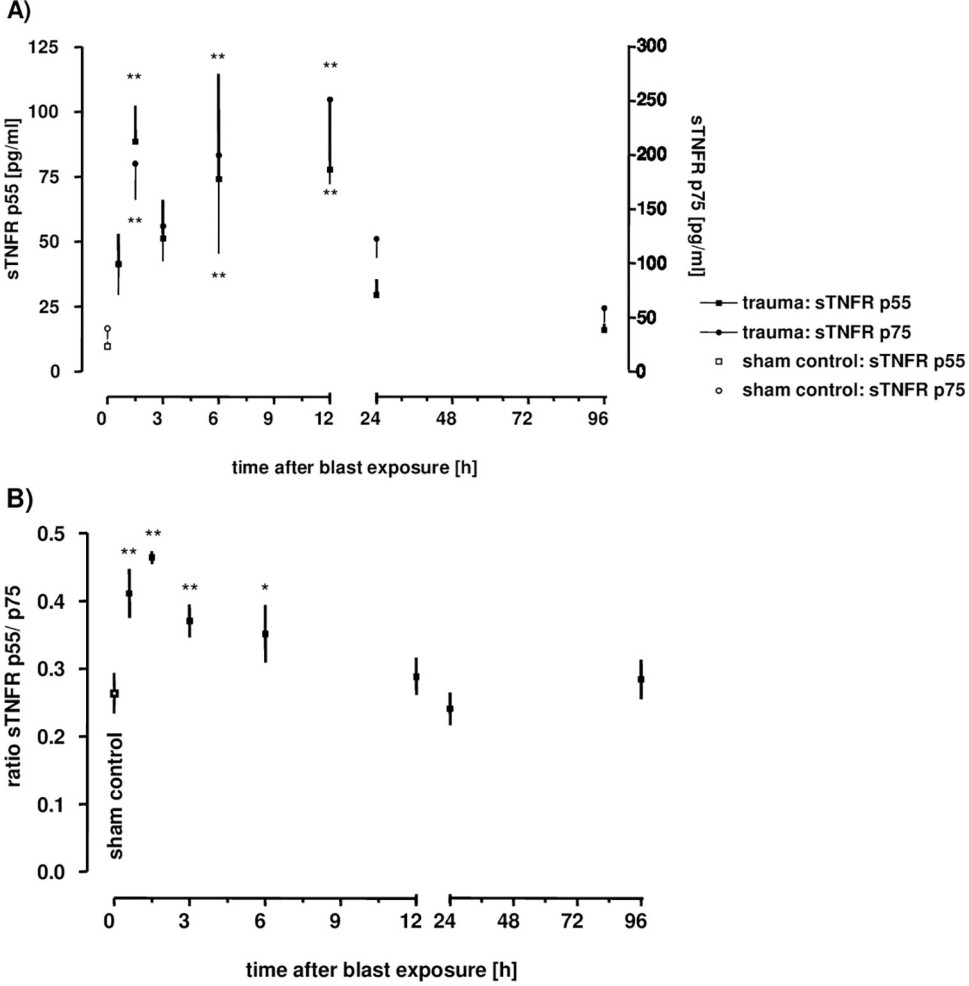

**Fig 10.** Time course of sTNFR p55 and p75 BAL concentrations in **A)** as well as the p55/ p75 ratio in **B)**. The animals were sacrificed 10 min (n = 6), 1.5 h (n = 3), 3 h (n = 3), 6 h (n = 3), 12 h (n = 4), 24 h (n = 3) or 96 hours (n = 3) after blast exposure, non-traumatized animals served as sham controls (n = 9). Levels of sTNFR p55 and p75 were assessed in BALFs. Data were analysed by one-way ANOVA and Dunnett's multiple comparison test: $^{*}$p $<$0.05 and $^{**}$p $<$0.01 vs. sham control. Unpaired t-test was performed to compare differences in p55 and p75 levels at different time-points: $^{**}$p = 0.0074 for p75 vs. p55 and $^{*}$p = 0.04 for p75 versus p55 at 24 hours and 96 hours after blast, respectively. Data are means ± SEM, number of experiments (n) in parentheses. Linear regression between p55 and p75 release revealing a Pearson r of r = 0.96; p $<$0.0001.

unique tissue architecture, is a blast-sensitive organ. Blast overpressure can physically destroy its vital compartments and induce inflammatory responses of varying degrees. Inflammatory responses are normally compartmentalised in the lungs, and the analysis of blood specimens provides an incomplete reflection of inflammatory events in the lungs [18]. As follows, we surveyed the presence or activity of mediators and inflammatory cells in the BALF of injured rat lungs versus sham controls in a defined set of blast-induced lung injuries in rats.

Obtained results demonstrated that following an initial blast-induced alveolar haemorrhage, the BAL red blood cell count, as well as the alveolar protein content, continuously reduced over time and both were no longer detectable after 96 h. This could be due to the elevated AMs amounts detected 96 h after the blast exposure. After haemorrhage, substantial quantities of haemoglobin or hemosiderin and ingested red blood cells in AMs can be detected. While peripheral blood leukocytes revealed no statistically significant differences

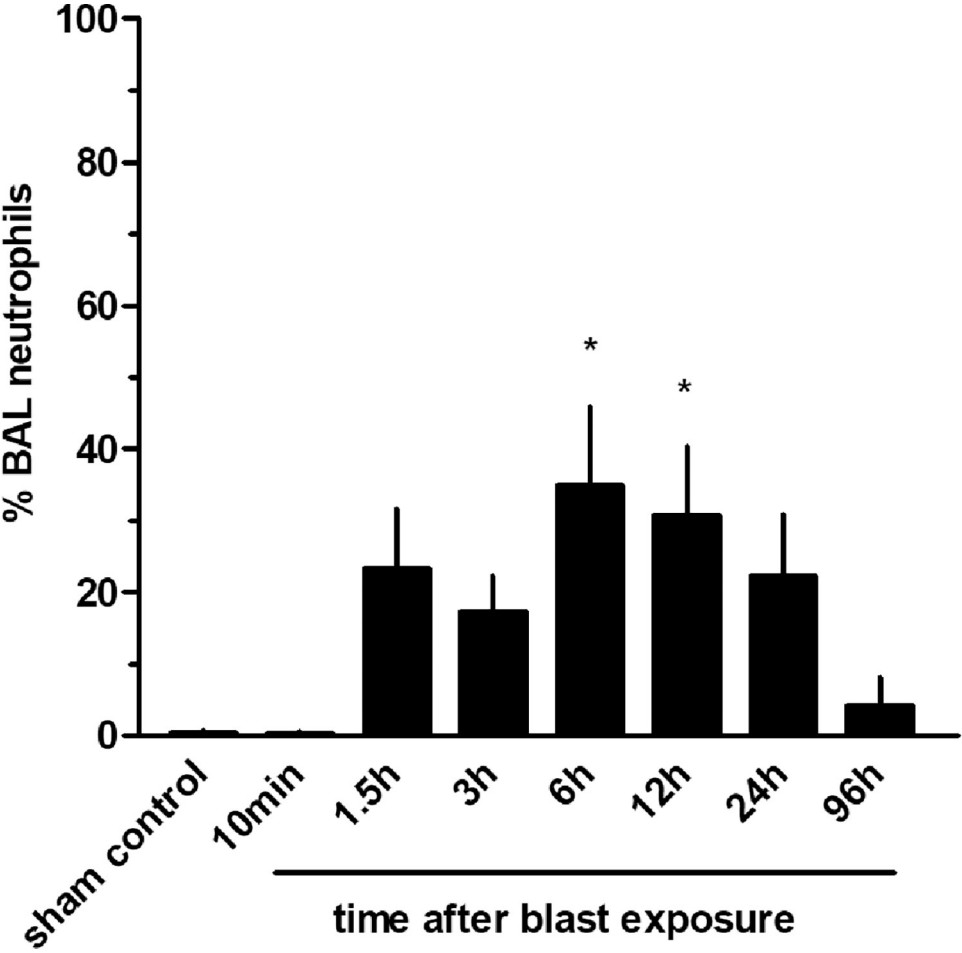

**Fig 11. Time-course of neutrophil (PMN) infiltration into the alveoli after thoracic trauma.** The animals were sacrificed 10 min (n = 6), 1.5 h (n = 3), 3 h (n = 3), 6 h (n = 3), 12 h (n = 4), 24 h (n = 3) or 96 hours (n = 3) after blast exposure, non-traumatized animals served as sham controls (n = 9). Statistical analysis was performed by one-way ANOVA and Dunnett's multiple comparison test: * p <0.05 vs. sham control. Data are expressed as mean ± SEM, number of experiments (n).

after the blast injury in our setting, the AMs' numbers increased starting 90 min after trauma and gained maximum level between 6 and 12 h in BAL. Significant protein accumulation in the alveolar space of traumatised lungs compared to sham controls confirms the disruption of the alveolar-interstitial and interstitial-alveolar barriers.

The LDH activity as a marker for cell damage varied in accordance with red blood cell counts and may therefore predominantly represent erythrocyte lysis; however, there is conceivably also cell death, apoptosis, or necrosis occurring in the alveolar septa, likely mostly including endothelial cells, interstitial cells, and alveolar type I cells. The increase in total protein content and alveolar haemorrhage together with an increase in the activity of LDH thus indicates both the disruption of the alveolar-capillary barrier and cell death. The concentration of eicosanoids in the BALFs revealed an immediate release after lung injury followed by a continuous decrease without a second increase. The inducible form of COX-2 could not be detected using RT-PCR in the BALFs within the first 6 h, suggesting that involvement of COX-2-dependent eicosanoids as mediators of a longer-standing inflammatory reaction is relatively unlikely.

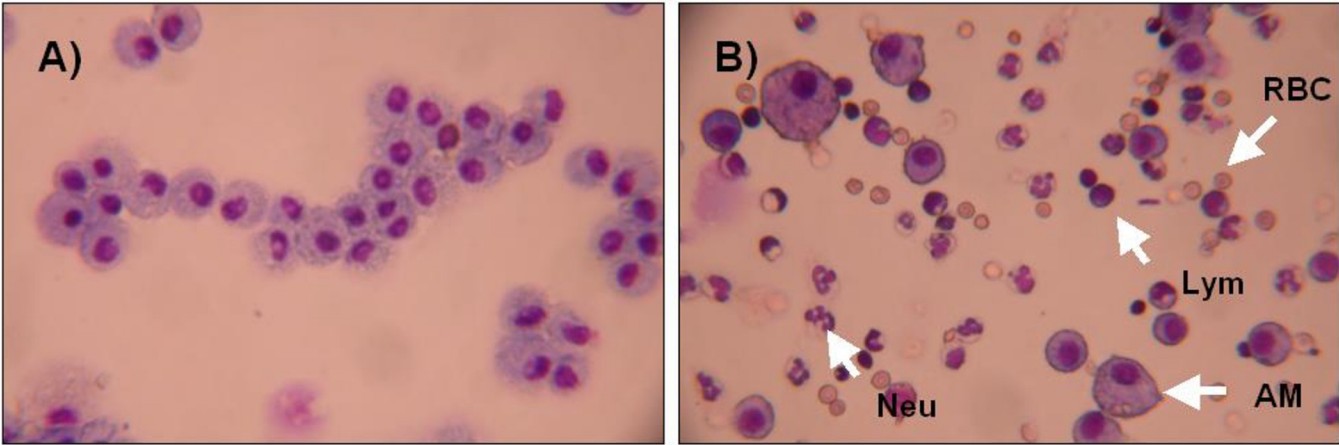

**Fig 12. Cytospin pictures after blast exposure. A)** sham control, **B)** 24 hours after blast exposure. **AM:** alveolar macrophages; **Neu:** neutrophils; **Lym:** lymphocytes; **RBC:** red blood cells.

MMP-9 was detectable in BALFs 6 h after blast exposure. MMP-9 is mostly expressed by inflammatory cells [19] and thus may be correlated with the observed neutrophil infiltration at this time. MMP-9 is a regulatory factor in neutrophil migration across the basement membranes [20] and involves equally in the degradation of the extracellular matrix [21]. The release and activation of MMP-2 were measurable within 24 h after the blast, whereas neither MMP-2 nor MMP-9 were detectable after 96 h and in BALF of non-traumatized rats. MMP-2 has been reported to be preferentially secreted by fibroblasts and epithelial cells [22] and may, therefore, be released after trauma-induced cell damage and participate in tissue remodelling.

**Table 1. Summary of time-dependent BAL findings compared to non-traumatized sham controls.**

| Read out | | Time course *in vivo* after blast injury | | | |
|---|---|---|---|---|---|
| | | peak/ minimum | maximal recovery | Conclusions | |
| Total protein content | ↑ | 1.5 h | 96 h | Disrupted alveolar-capillary barrier/cell death | **time-dependent recovery** |
| Alveolar haemorrhage | ↑ | 1.5 h | 96 h | | |
| LDH activity | ↑ | 10 min | 96 h | | |
| Wet/dry ratio | ↑ | 1.5 h | 6 h | Pulmonary infiltrates | |
| TxB₂ | ↑ | 10 min | 3 h | Early prostanoid release | |
| 6-keto PGF1α | ↑ | 10 min | 3 h | | |
| TNF | ↑ | 12 h | 96 h | Pro-inflammatory cytokine release | |
| IL-6 | ↑ | 6 h | 96 h | | |
| IL-10 | ↓ | 10 min | 96 h | Early decrease | |
| CINC-3 | ↑ | 6 h | 96 h | Pro-inflammatory chemokine release | |
| sTNFR p55/ p75 ratio | ↑ | 10 min | 24 h | Anti-inflammatory counterregulation | |
| Neutrophil infiltration | ↑ | 6–12 h | > 96 h | Inflammation | |
| MMP-2 | ↑ | 10 min | 96 h | | |
| MMP-9 | ↑ | 6 h | 96 h | | |
| MPO | ↑ | 12 h | 96 h | | |

↑ Upregulation; ↓ Downregulation; **CINC-1:** cytokine-induced neutrophil chemoattractants; **IL-6:** Interleukin 6; **IL-10:** Interleukin 10; **LDH:** lactate dehydrogenase activity; **MMP-2:** Matrix Metallopeptidase 2; **MMP-9:** Matrix Metallopeptidase 9; **MPO:** myeloperoxidase; **sTNFR:** Soluble tumour necrosis factor receptors; **TNF:** tumour necrosis factor; **TXB2:** thromboxane B2.

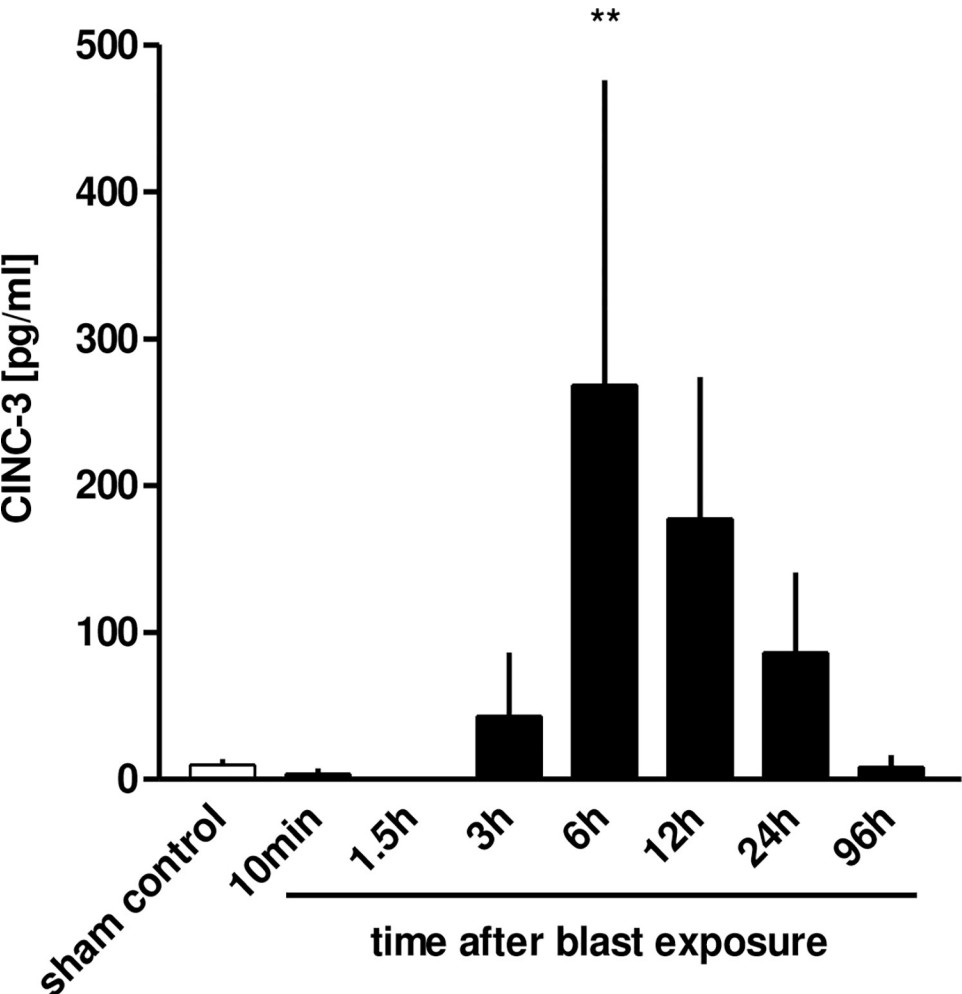

**Fig 13. CINC-3 measurement in the BALF of traumatized rat lungs at different time-points after blast exposure:**
**10 min (n = 6), 1.5 h (n = 3), 6 h (n = 3), 12 h (n = 4), 24 h (n = 3), 24 h (n = 3) and 96 h (n = 3).** Non-traumatized
animals served as sham controls (n = 9). Statistical analysis was performed by one-way ANOVA (p = 0.0014) and
Dunnett's multiple comparison test: **p <0.01 vs. sham control. Data are expressed as mean ± SEM, number of
experiments (n).

6 h after trauma, IL-6 levels in BALF were found significantly elevated, as compared to
non-traumatised control lungs and consequently returned back to their baseline value between
12 and 96 h after insult. This preliminary increase is in accordance with plasma IL-6 levels
obtained from polytrauma patients [4, 23]. IL-6 generation is provoked at least partly by stimu-
lation caused by TNF and IL-1β, and it has been proposed that IL-6 integrates early signals
produced in the inflammatory response [24]. Some data support the concept that the produc-
tion of TNF and IL-6 is regulated independently [25]. A crucial anti-inflammatory role for IL-
6 through controlling the levels of other pro-inflammatory cytokines [26] and inducing the
formation and release of sTNFR have been proposed [27]. IL-6 has been reported to be a
marker of the intensity of the injury, as their local peak after blunt chest trauma coincided
with the systemic inflammatory response accompanied by an impaired function of the pulmo-
nary endothelial barrier [28] bearing some prognostic value for the survival in the preliminary
posttraumatic phase [23].

In the current model, in addition to comparison to non-traumatized control lungs, the BAL TNF levels of traumatized lungs were significantly elevated from 3 h after trauma on and returned to basal values after 96 h. In a similar study with rats, up to a 10 fold BAL TNF level at 6 and 24 h after the blast exposure have been reported [29].

sTNFRs, even at low TNF concentrations, can affect TNF functions [30]. It is assumed that shedding of sTNFRs in response to high TNF concentrations could inhibit TNF activity and thus localise inflammatory response [31]. Moreover, disabling TNF proinflammatory action might facilitate anti-inflammatory action of TNF, which is mediated by its lectin-like domain [32–34]. We observed elevated sTNFR1 and 2 (p55 and p75) levels in rat lung BALFs within 90 minutes after trauma, analogous to plasma sTNFR levels of polytrauma patients [4, 35], suggesting that thorax trauma probably triggers the rapid release of receptors on the cell surface of injured lung tissue or activated leukocytes. Similar to the measured BAL TNF levels, BAL sTNFR1 and 2 levels were comparatively low as compared to plasma sTNFR1 and 2 levels observed in trauma patients [4, 36, 37]. Part of it may be explained by dilution due to the BAL, which is probably being about 100-fold [24]. In contrast to resembling studies [4, 36], we observed that receptor levels returned to non-traumatized control levels after 96 h, indicating a peak profile of sTNFR rather than a sustained elevation. Clinical studies reported decline of sTNFR levels in patients who recovered from injury [35], and accordingly the decrease in BAL sTNFR seems related to biologic recovery from primary blast injury. The correlation between each TNFRs with the IL-6 release within 96 h after blast supports the hypothesis that IL-6 may be involved in the regulation of both sTNFRs. Elevated sTNFR1 levels in cancer patients after administration of recombinant IL-6 have been also observed [27]. In the present study, the BAL levels of both sTNFRs were elevated within 90 minutes after trauma, which is in agreement with severely injured patients described previously [38]. An early shift towards sTNFR p55 release has been similarly reported [4]. Proteases and IL-6 have been discussed as the factors that may be responsible for the increased or prolonged release of sTNFR [27]. Neutrophil-derived proteases have been shown to be elevated, predominantly in patients with isolated thorax trauma [23]. The involvement of locally produced IL-6 in the early appearance of sTNFR subtypes in the injured lung after primary blast injury might be speculated. Whether sTNFR levels in BALFs represents the response to local TNF production and/ or the reason why BAL TNF levels were relatively moderate and plasma levels were even not measurable, cannot be explained by the present study.

At all time points studied up to 96 h after trauma, plasma TNF was below the detection limit, neither measurable by ELISA, nor functionally by the WEHI-bioassay. This underlines that TNF is mainly produced and secreted locally in the alveolar tissue, or by AMs. According to the biological importance of TNF, in ARDS patients on the first day, non-survivors had significantly higher BAL plasma ratios for TNF, IL-1β, IL-6, and IL-8, but also over time, BAL plasma ratios for TNF, IL-1ß, and IL-6 remained elevated in non-survivors, while they decreased in survivors [39].

IL-10 is an anti-inflammatory cytokine being produced by AMs [40]. IL-10 is assumed as a counter-regulatory cytokine that inhibits cytokine production by stimulated macrophages [41], is detectable in BALFs of ARDS patients in comparatively low concentrations (10–20 pg/ml) [24] and even significantly decreased levels over 24 h after blast exposure as compared to non-traumatized control lungs baseline levels, and normalized after 96 h. In the study of Liener *et al.* (2011), IL-10 as an indicator of anti-inflammatory regulation exhibited a diminishing tendency, compared to shame control [40]. Low IL-10 levels may introduce the function of this cytokine in the alveolar environment. In contrast, Hensler *et al.* (2002) found elevated plasma IL-10 values within 3 h after trauma in patients suffering from multiple injuries [4]. The lung perpetuates a constitutive level of anti-inflammatory cytokine IL-10, which is not

enhanced after aerosol exposure of endotoxin, whereas systematically, intraperitoneal delivery of endotoxin resulted in an increase in circulating IL-10 [26]. Initiation of coagulation by tissue factor (TF) has been shown to be a powerful regulator of local inflammatory responses [42]. Accordingly, blockade of Tissue Factor-Factor VIIa complex protected the lung from injury by LPS in part by reducing local expression of pro-inflammatory cytokines and local elaboration of IL-1β, IL-6, and IL-10 [43]. Furthermore, trauma induces TF expression on monocytes have been discovered [44], which might be one reason for the observed decline in BAL IL-10 after blast injury in this model.

Thoracic trauma increased neutrophil chemokine production as shown in BAL with Cytokine-induced neutrophil chemoattractant 3 (CINC-3, MIP-2, GRO beta, or CXCL2), a member of the CXC subfamily of chemokines, peaking in the alveolar space about at 6 h. AMs are thought to be the major source of α-chemokines such as CINC-3 in the airspaces [24, 45] and produce CINC e.g. in response to LPS *in vitro* [46] and *in vivo* [45, 47, 48]. CINC-3 has been reported to play the most important role in the recruitment of neutrophils and their activation during lipopolysaccharide- mediated ALI [47]. Lung injury by large tidal volume ventilation demonstrated increased IL-8 levels in BALF originating mainly from neutrophils accumulated in the lung [49]. In chronic inflammation induced by experimental pulmonary silicosis, CNC-3 was shown to be a chemoattractant in the formation of granulomas [50]. In contrast, the level of CINC-3 was shown to be low in experimental chronic bronchopulmonary infections of rats with *Pseudomonas aeruginosa* [51], and after experimental airway exposure of rats to staphylococcal enterotoxin [52] indicating the important role of the causative agent damaging the lung tissue and initiating the inflammatory response concerning CINC-3.

The reduced wet/dry ratio assessed 3 and 6 h after the blast, as compared to the immediately assessed lungs can be related to both reduced oedema generation and elevated oedema resorption. Whether these effects are related to possible endogenous epinephrine-induced actions or to actions due to the lectin-like tip region of TNF cannot be explained as we did not specifically attempt to block possible trauma-related epinephrine or TNF lectin-like region effects. These results suggest that sodium channels are at least in part functional after blast injury in this model [33, 34]. Thus, increased alveolar fluid clearance might be an underlying mechanism for the postulated fluid resorption *in vivo* within the first six hour after trauma. Indeed the rise in wet/dry ratio at 15 minutes is solely due to intra-alveolar haemorrhage, while the rise of this ratio at 3 hours could be due to both haemorrhage and oedema.

The model and the presented results are associated with many limitations. The *in vivo* model is a rather standardized severe trauma model in which about a tenth of rats died, and is therefore done with minimal numbers of animals per group or time point and thus with restricted biosamples. Thoracic trauma deteriorates lung function, leading to a certain degree of respiratory failure, which has mainly been indirectly shown here as e.g. in terms of wet-dry ratio. The effect of anaesthetics like halothane and the opioid buprenorphine cannot be excluded, but also pain might be a potential modifier of outcome. In the experimental setting the use of sham controls may best correct for those effects. The present study established a blast-induced lung injury model in rats, which provides rather valid and reproducible results. The experimental data complement those from human patients and may once give avenues to modulating injury, which is of importance in view of the impact of trauma on all, but especially also on young patients and thus on society. In addition to the time-dependent resolution of the trauma-related alveolar protein-rich fluid, these results gave evidence of the high and rapid repair including the alveolar fluid, cell, and cell debris clearance capacity of the lung. Furthermore, it is unclear whether shorter and possibly more representative duration blast over pressure related injuries are distinct from longer and more frequent duration blast over pressure related injuries.

## Supporting information

**S1 Fig. Timeline of the experiment.**
(PPT)

**S1 File. The ARRIVE essential 10.**
(DOC)

## Acknowledgments

This work has mainly been performed by Dr. Katja Eichert as work on her thesis; all authors highly acknowledge her work for this paper. In addition, we acknowledge the collaboration of Ulrich Liener, Markus Knöferl, Florian Gebhard, Lothar Kinzl (Univ. of Ulm), Manfred Kind (Department of Pathology, Konstanz), and the technical assistance of Bruno Erne (Technical Dept., Univ. of Konstanz), Clemens Braun, Anne Hildebrandt and Sandra Unger (Univ. of Konstanz).

## Author Contributions

**Conceptualization:** Jürg Hamacher, Hanno Huwer.

**Data curation:** Jürg Hamacher, Hanno Huwer.

**Formal analysis:** Jürg Hamacher, Rudolf Lucas.

**Funding acquisition:** Jürg Hamacher.

**Investigation:** Jürg Hamacher, Uz Stammberger, Rudolf Lucas.

**Methodology:** Jürg Hamacher, Hanno Huwer, Uz Stammberger, Rudolf Lucas.

**Project administration:** Yalda Hadizamani.

**Resources:** Jürg Hamacher.

**Software:** Jürg Hamacher, Hanno Huwer.

**Supervision:** Jürg Hamacher, Uz Stammberger, Rudolf Lucas.

**Validation:** Jürg Hamacher, Uz Stammberger, Albrecht Wendel, Rudolf Lucas.

**Visualization:** Jürg Hamacher.

**Writing – original draft:** Jürg Hamacher, Hanno Huwer.

**Writing – review & editing:** Jürg Hamacher, Yalda Hadizamani, Ueli Moehrlen, Lia Bally, Uz Stammberger, Rudolf Lucas.

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
