## [Decision Letter · Decision Letter 0]

4 Jul 2022

PONE-D-22-09944Characteristics of inflammatory response and repair after experimental blast lung injury in ratsPLOS ONE

Dear Dr. Hadizamani,

Thank you for submitting your manuscript to PLOS ONE. After careful consideration, we feel that it has merit but does not fully meet PLOS ONE’s publication criteria as it currently stands. Therefore, we invite you to submit a revised version of the manuscript that addresses the points raised during the review process.

We look forward to receiving your revised manuscript.

Kind regards,

Nicolas Tsapis, Ph.D.

Academic Editor

PLOS ONE

Journal Requirements:

3. As part of your revision, please complete and submit a copy of the Full ARRIVE 2.0 Guidelines checklist, a document that aims to improve experimental reporting and reproducibility of animal studies for purposes of post-publication data analysis and reproducibility: https://arriveguidelines.org/sites/arrive/files/Author%20Checklist%20-%20Full.pdf (PDF). Please include your completed checklist as a Supporting Information file. Note that if your paper is accepted for publication, this checklist will be published as part of your article.

4. Thank you for submitting the above manuscript to PLOS ONE. During our internal evaluation of the manuscript, we found significant text overlap between your submission and the following previously published works, some of which you are an author.

https://d-nb.info/969514263/34

Please revise the manuscript to rephrase the duplicated text, cite your sources, and provide details as to how the current manuscript advances on previous work. Please note that further consideration is dependent on the submission of a manuscript that addresses these concerns about the overlap in text with published work.

6. Thank you for stating the following in the Competing Interests section: 

"The work of YH, UM, and JH have been funded by Lungen- und Atmungsstiftung Bern, Switzerland which provides non-restricted financial support toward research on lung, respiratory and sleep-related respiratory diseases as well as related rehabilitation and lifestyle change issues such as exercise, smoking cessation, etc. JH is the chair of Lungen- und Atmungsstiftung Bern and his position did not influence the design of the study, the collection of the data, the analysis or interpreta-tion of the data, the decision to submit the manuscript for publication, or the writing of the manuscript and did not present any financial conflicts. Also, the work of JH was supported by a grant from the Deutsche Forschungsgemeinschaft (FOR 321/2-1; research group “Endogenous tissue injury: Mechanisms of autodestruction”) and by the Herrmann Josef Schieffer Prize of the “Freunde des Universitätsklinikums Homburg e.V.”. All remaining authors (HH, MM, LB, US, AW and RL) declare no potential financial or non-financial conflict of interest with the work presented here."

7. PLOS ONE now requires that authors provide the original uncropped and unadjusted images underlying all blot or gel results reported in a submission’s figures or Supporting Information files. This policy and the journal’s other requirements for blot/gel reporting and figure preparation are described in detail at https://journals.plos.org/plosone/s/figures#loc-blot-and-gel-reporting-requirements and https://journals.plos.org/plosone/s/figures#loc-preparing-figures-from-image-files. When you submit your revised manuscript, please ensure that your figures adhere fully to these guidelines and provide the original underlying images for all blot or gel data reported in your submission. See the following link for instructions on providing the original image data: https://journals.plos.org/plosone/s/figures#loc-original-images-for-blots-and-gels. 

8. We note that you have included the phrase “data not shown” in your manuscript. Unfortunately, this does not meet our data sharing requirements. PLOS does not permit references to inaccessible data. We require that authors provide all relevant data within the paper, Supporting Information files, or in an acceptable, public repository. Please add a citation to support this phrase or upload the data that corresponds with these findings to a stable repository (such as Figshare or Dryad) and provide and URLs, DOIs, or accession numbers that may be used to access these data. Or, if the data are not a core part of the research being presented in your study, we ask that you remove the phrase that refers to these data.

Reviewers' comments:

Reviewer's Responses to Questions

**Comments to the Author**

1. Is the manuscript technically sound, and do the data support the conclusions?

Reviewer #1: Yes

Reviewer #2: Partly

2. Has the statistical analysis been performed appropriately and rigorously? 

Reviewer #1: N/A

Reviewer #2: No

3. Have the authors made all data underlying the findings in their manuscript fully available?

Reviewer #1: Yes

Reviewer #2: No

4. Is the manuscript presented in an intelligible fashion and written in standard English?

Reviewer #1: Yes

Reviewer #2: No

5. Review Comments to the Author

Reviewer #1: The article by Hamacher et al. concerns the characterization of inflammatory response an repair after experimental blast injury.

They showed that following blast injury an alteration of cytokines and activity cells is observed. They showed a neutrophil infiltration into the alveolar space, signature of inflammation.

The study is well conduct but I would like to mention that I am not specialized in blast injury.

I have some comments.

Figure 2 is is not mentioned in the text as figure 3 (results paragraph e3.2, Figure 3C is mentioned instead of figure 2).

What is the number of animals per group? 1?

n is the number of experiments ? one experiment is one animal /group ?

Regarding the statistics, I am quite surprised because the results are expressed as an mean +/- SEM. SEM is important. It would be interesting to represent the results as a scatter plot to show the individual response of all animals.

For figure 11, the points are connected, which is not acceptable because the animals are not the same for each time. This figure must be modified and represented like the others.

I suggest to include the MPO result with neutrophils paragraph.

For the MMP part (results) it could be shorter by extracting a part already integrated in the discussion.

The discussion is rather long and not concise enough. It's confusing. Why not discuss the elements as a whole. For example, the discussion on TNF is in three paragraphs or even 4 and quite long and a bit redundant. PMN, MMP, CINC could be discussed together in sub-paragraphs.

Reviewer #2: This manuscript dealing with post blast lung inflammatory response presents results of interest for the scientific community. However, numerous issues prevent this manuscript being published as is.

Global remarks:

1- the manuscript has to be professionaly language edited

2- the figure's numbers are incorrect

3- citations are sometimes not cited correctly

Specific remarks:

4- the statistics are using parametric tests with very small n in each groups. Non parametric tests need to be used. Also, when using small groups, data must be presented as Median [IQR] instead of mean (SEM or SD)

5- the authors does not discuss the similarities/differences with a classical blunt trauma model. An interesting question should be: How the blast related lung injuries are different from blunt chest trauma related lung injuries?

6- thus, the definition of the incoming blast overpressure wave is of importance. The OP wave provided by the shock tube is very short (0.6 ms) where standout open field OP waves are close to 2-4 ms. It is understandable that even if very short, this OP wave is responsible for lung injuries, but this singularity is not discussed in the paper. how short OP wave related lung injury is different from normal length OP wave related lung injury?

7- the wet to dry ratio study is of interest but it does not take the intra-alveolar hemorrhage into account. Also, methods don't describe if W/D ratio lungs were previously injected for BAL... The methods need to be clearer.

8- concerning the methods, why is the anesthetic strategy biphasic (firs halothan then pentothal) ? Also the pro-inflammatory effect of buprenorphine is not discussed.

6. PLOS authors have the option to publish the peer review history of their article (what does this mean?). If published, this will include your full peer review and any attached files.

Reviewer #1: No

Reviewer #2: **Yes: **Nicolas J. Prat

---

## [Author Response · Author response to Decision Letter 0]

15 Sep 2022

5. Review Comments to the Author

Reviewer #1: The article by Hamacher et al. concerns the characterization of inflammatory response an repair after experimental blast injury.They showed that following blast injury an alteration of cytokines and activity cells is observed. They showed a neutrophil infiltration into the alveolar space, signature of inflammation. The study is well conduct but I would like to mention that I am not specialized in blast injury.

I have some comments.

1. Figure 2 is not mentioned in the text as figure 3 (results paragraph e3.2, Figure 3C is mentioned instead of figure 2).

The reviewer is right, now we have corrected figures’ numbers.

2. What is the number of animals per group? 1? n is the number of experiments ? one experiment is one animal /group ?

In every paragraph the numbers of animals per experiment has been indicated.

3. Regarding the statistics, I am quite surprised because the results are expressed as an mean +/- SEM. SEM is important. It would be interesting to represent the results as a scatter plot to show the individual response of all animals.

Unfortunately we have not the original data any more in our hands and are therefore not able to re-plot the graphs in scatter plots. We asked the university of Konstanz to have access to the original data, but have not yet got this access as all those laboratory books including all data from assays are given in those books. Unfortunately so far we have not had this access. 

4. For figure 10, the points are connected, which is not acceptable because the animals are not the same for each time. This figure must be modified and represented like the others.

Reviewer is totally right. We have modified this figure.

5. I suggest to include the MPO result with neutrophils paragraph.

Thank you for this comment. It has been done.

For the MMP part (results) it could be shorter by extracting a part already integrated in the discussion.

Thank you for this comment. Now, it has been done.

6. The discussion is rather long and not concise enough. It's confusing. Why not discuss the elements as a whole. For example, the discussion on TNF is in three paragraphs or even 4 and quite long and a bit redundant. PMN, MMP, CINC could be discussed together in sub-paragraphs.

Reviewer #2: This manuscript dealing with post blast lung inflammatory response presents results of interest for the scientific community. However, numerous issues prevent this manuscript being published as is.

Global remarks:

1- the manuscript has to be professionally language edited

We acknowledge that Dr. Rudolf Lucas who works since > 15 y in the USA co-wrote and several times revised the text and that we therefore think that language is acceptable.

2- the figure's numbers are incorrect

The reviewer is totally right. Now the figure’s numbersa have been corrected.

3- citations are sometimes not cited correctly

Thank you for this comment. They have been corrected and corrected.

Specific remarks:

4- the statistics are using parametric tests with very small n in each groups. Non parametric tests need to be used. Also, when using small groups, data must be presented as Median [IQR] instead of mean (SEM or SD)

We always were attentive to check whether we had to do non-parametric testing. In all the dataset we are confident that we did not violate the rules for parametric tests. We otherwise would have given the data and results with median and ranges and would have indicated the non-parametric test methods. 

5- the authors does not discuss the similarities/differences with a classical blunt trauma model. An interesting question should be: How the blast related lung injuries are different from blunt chest trauma related lung injuries?

The reviewer is right that we did not discuss. We discussed it in the discussion section as: Blast thoracic injury seems to be the most dangerous type of non-penetrating thoracic injury that may more regularly lead to death due to acute lung injury than in other non-penetrating thoracic injuries (CDC Explosions and blast injuries. A primer for Clinicians). 

6- thus, the definition of the incoming blast overpressure wave is of importance. The OP wave provided by the shock tube is very short (0.6 ms) where standout open field OP waves are close to 2-4 ms. It is understandable that even if very short, this OP wave is responsible for lung injuries, but this singularity is not discussed in the paper. how short OP wave related lung injury is different from normal length OP wave related lung injury?

The question is for sure of physical and of pathophysiological importance. We had a clear and defined experimental setting, also certainly with minimal distance to the subject, and we have no clear clue to alternative and slower overpressure waves. Unfortunately, we can therefore not answer the question. 

7- the wet to dry ratio study is of interest but it does not take the intra-alveolar hemorrhage into account. Also, methods don't describe if W/D ratio lungs were previously injected for BAL... The methods need to be clearer.

The wet to dry ratio is a standardized procedure performed in acute lung injury research since decades. Every clinically relevant acute lung injury is accompanied with some degree of destruction of the alveolar-capillary barrier and therefore of alveolar haemorrhage. This is e.g. very well seen in the scientific work on clinical acute lung injury and ARDS work where bronchoalveolar lavages have been analysed. The only point is that there it is in many times not quantified, but it is present. Therefore in virtually all data on severe acute lung injury some alveolar haemorrhage is present. We therefore are confident that this does not inferiorize the dataset presented on an injury where about 10 % of rats died within a few minutes thereafter. We consider it as a normal phenomenon not disturbing the value of wet to dry ratio. 

8- Concerning the methods, why is the anesthetic strategy biphasic (firs halothan then pentothal) ? Also the pro-inflammatory effect of buprenorphine is not discussed.

The anaesthetic strategy was also given by the ethics committee, as in animal research the main focus is - e. g. besides reduction of animal numbers – to give the animals intravitally minimal harm. We therefore had to cover the whole intravital research for the rats with maximal analgesia, and therefore this anaesthetic strategy had to be chosen. Neither pain or dyspnea itself, nor an analgetic drug is probably completely free of biologic effects, and therefore any way with more or less pain and with any major analgetic drug may modify to a certain extent the read-outs as e.g. discussed by you. 

This dilemma of animal in vivo research is in many instances best solved with rather generous analgetics and anaesthetics, and with the use of sham controls, as it is done only with very few scientifically and ethically acceptably exceptions in the animal research and its moral obligation of humane care. 

The literature on buprenorphine is impressive and shows a number of anti-as a number of prl-inflammatory research papers. 

We integrated in the limitation section one sentence on the possible effect of buprenorphine concerning the influence on outcome.

---

## [Decision Letter · Decision Letter 1]

6 Dec 2022

PONE-D-22-09944R1Characteristics of inflammatory response and repair after experimental blast lung injury in ratsPLOS ONE

Dear Dr. Hadizamani,

Thank you for submitting your manuscript to PLOS ONE. After careful consideration, we feel that it has merit but does not fully meet PLOS ONE’s publication criteria as it currently stands. Therefore, we invite you to submit a revised version of the manuscript that addresses the points raised during the review process.

We look forward to receiving your revised manuscript.

Kind regards,

Nicolas Tsapis, Ph.D.

Academic Editor

PLOS ONE

Journal Requirements:

Reviewers' comments:

Reviewer's Responses to Questions

**Comments to the Author**

1. If the authors have adequately addressed your comments raised in a previous round of review and you feel that this manuscript is now acceptable for publication, you may indicate that here to bypass the “Comments to the Author” section, enter your conflict of interest statement in the “Confidential to Editor” section, and submit your "Accept" recommendation.

Reviewer #2: (No Response)

2. Is the manuscript technically sound, and do the data support the conclusions?

Reviewer #2: Yes

3. Has the statistical analysis been performed appropriately and rigorously? 

Reviewer #2: Yes

4. Have the authors made all data underlying the findings in their manuscript fully available?

Reviewer #2: Yes

5. Is the manuscript presented in an intelligible fashion and written in standard English?

Reviewer #2: Yes

6. Review Comments to the Author

Reviewer #2: We thank the authors for their responses, however some points needs to be clarified in the manuscript.

Point 1-3: ok

Point 4 – Statistics.

The authors state in their response and in the ARRIVE document that they did all the controls needed to use the parametric tests. This have to be also stated in the manuscript, section 2.6.

Point 5: Comparison with blunt thoracic trauma.

The authors add a comparison with blunt trauma concerning the general severity. The manuscript would benefit from an addition on the difference, even if poorly known, on the pathophysiological aspect of the lung contusion formation between the two different sorts of insult.

Point 6: Discussion on OP characteristics

In their response, the authors acknowledge they cannot answer the question whether short duration blast OP related injuries are or are not different from longer (and more representative) duration BOP related injuries. This has to be discussed in the manuscript as a limitation.

Point 7: W/D ratio

The information given in the response should benefit to the discussion in the manuscript.

For sure, the rise in W/D ration at 15 min is solely due to intra-alveolar hemorrhage, the rise at 3h due to both hemorrhage and edema.

Point 8: OK

New point: Can the authors specify at what time control animals were sacrificed?

7. PLOS authors have the option to publish the peer review history of their article (what does this mean?). If published, this will include your full peer review and any attached files.

Reviewer #2: **Yes: **Nicolas J. PRAT

---

## [Author Response · Author response to Decision Letter 1]

13 Jan 2023

Response to Reviewers

6. Review Comments to the Author

Reviewer #2: We thank the authors for their responses, however some points needs to be clarified in the manuscript.

Point 1-3: ok

Point 4 – Statistics.

The authors state in their response and in the ARRIVE document that they did all the controls needed to use the parametric tests. This have to be also stated in the manuscript, section 2.6.

Thank you very much for this comment. In the section 2.6., and ARRIVE document we mentioned that "Notably, we have done all the controls needed to use the parametric tests."

Point 5: Comparison with blunt thoracic trauma. The authors add a comparison with blunt trauma concerning the general severity. The manuscript would benefit from an addition on the difference, even if poorly known, on the pathophysiological aspect of the lung contusion formation between the two different sorts of insult.

The shock wave produced by explosions and high-velocity projectiles may cause serious trauma to the pulmonary parenchyma. The onset of damage from a blast wave occurs when the blast wave hits the thoracic wall and compresses thorax and intrathoracic tissues and thus, in particular, the intrathoracic gas volume. The resulting forces exceed the tensile strength of the tissues and damage their structural texture. So the most common injury associated with a blast wave is a pulmonary contusion. On microscopic examinations the changes closely resemble contusions in non-penetrating blunt chest trauma.

Lung injury by a blast wave occurs when an overpressure of about 40 PSI is reached. Blunt chest trauma leads to a comparable lung contusion, when traumata produce instantaneous changes in velocity when frontal crashes into a fixed object occur as well as near-side lateral impact during a vehicular crash (O'Connor et al., 2009).

Pulmonary contusion from blunt non-penetrating chest trauma is often accentuated to one side and in areas where the thoracic cage is relatively elastic and compressible. Radiographic changes are localized accordingly. Explosion-related trauma, on the other hand, often produces a characteristic "butterfly" pattern in chest CT due to the broad front of the blast wave and reflections of the blast wave if explosions occur in an enclosed space.

We have added in the introduction this interesting point for the reader, and we thank the reviewer for this input. We discussed this subject in less detail, so that we did not too much extend the text. 

Point 6: Discussion on OP characteristics

In their response, the authors acknowledge they cannot answer the question whether short duration blast OP related injuries are or are not different from longer (and more representative) duration BOP related injuries. This has to be discussed in the manuscript as a limitation.

Thank you very much for noting this issue. We formulated this fact in the limitation section as follows: Furthermore, it is unclear whether shorter and possibly more representative duration blast over pressure related injuries are distinct from longer and more frequent duration blast over pressure related injuries.

Point 7: W/D ratio

The information given in the response should benefit to the discussion in the manuscript.For sure, the rise in W/D ration at 15 min is solely due to intra-alveolar hemorrhage, the rise at 3h due to both hemorrhage and edema.

I appreciate your comment in this case. In the related section now we have noted that: Indeed the rise in wet/dry ratio at 15 minutes is solely due to intra-alveolar haemorrhage, while the rise of this ratio at 3 hours could be due to both haemorrhage and oedema.

Point 8: OK

New point: Can the authors specify at what time control animals were sacrificed?

10 minutes, which is the identical time point as the sacrifice of the first group concerning time points.

Reference

O'Connor, J. V., Kufera, J. A., Kerns, T. J., Stein, D. M., Ho, S., Dischinger, P. C., & Scalea, T. M. (2009). Crash and occupant predictors of pulmonary contusion. J Trauma, 66(4), 1091-1095. https://doi.org/10.1097/TA.0b013e318164d097

---

## [Editor Report · Decision Letter 2]

24 Jan 2023

Characteristics of inflammatory response and repair after experimental blast lung injury in rats

PONE-D-22-09944R2

Dear Dr. Hadizamani,

We’re pleased to inform you that your manuscript has been judged scientifically suitable for publication and will be formally accepted for publication once it meets all outstanding technical requirements.

Kind regards,

Nicolas Tsapis, Ph.D.

Academic Editor

PLOS ONE

Additional Editor Comments (optional):

The authors have addressed all the criticism raised by the reviewers, have answered appropriately and have modified the article appropriately. In my opinion the article can be accepted.
---

## [Editor Report · Acceptance letter]

8 Mar 2023

PONE-D-22-09944R2 

Characteristics of inflammatory response and repair after experimental blast lung injury in rats 

Dear Dr. Hadizamani:

I'm pleased to inform you that your manuscript has been deemed suitable for publication in PLOS ONE. Congratulations! Your manuscript is now with our production department. 

Kind regards, 

on behalf of

Dr. Nicolas Tsapis 

Academic Editor

PLOS ONE